# Nitrogen Starvation-Responsive MicroRNAs Are Affected by Transgenerational Stress in Durum Wheat Seedlings

**DOI:** 10.3390/plants10050826

**Published:** 2021-04-21

**Authors:** Haipei Liu, Amanda J. Able, Jason A. Able

**Affiliations:** Waite Research Institute, School of Agriculture, Food & Wine, The University of Adelaide, Urrbrae, SA 5064, Australia; amanda.able@adelaide.edu.au (A.J.A.); jason.able@adelaide.edu.au (J.A.A.)

**Keywords:** nitrogen starvation, water-deficit and heat stress, transgenerational effects, cross stress tolerance, microRNAs, crop improvement

## Abstract

Stress events have transgenerational effects on plant growth and development. In Mediterranean regions, water-deficit and heat (WH) stress is a frequent issue that negatively affects crop yield and quality. Nitrogen (N) is an essential plant macronutrient and often a yield-limiting factor for crops. Here, the response of durum wheat seedlings to N starvation under the transgenerational effects of WH stress was investigated in two genotypes. Both genotypes showed a significant reduction in seedling height, leaf number, shoot and root weight (fresh and dry), primary root length, and chlorophyll content under N starvation stress. However, in the WH stress-tolerant genotype, the percentage reduction of most traits was lower in progeny from the stressed parents than progeny from the control parents. Small RNA sequencing identified 1534 microRNAs in different treatment groups. Differentially expressed microRNAs (DEMs) were characterized subject to N starvation, parental stress and genotype factors, with their target genes identified in silico. GO and KEGG enrichment analyses revealed the biological functions, associated with DEM-target modules in stress adaptation processes, that could contribute to the phenotypic differences observed between the two genotypes. The study provides the first evidence of the transgenerational effects of WH stress on the N starvation response in durum wheat.

## 1. Introduction

Environmental stresses such as water deficiency, extreme temperatures, and soil nutrient deficiency present significant challenges to the development and production of crops. Durum wheat (*Triticum turgidum* L. ssp. *durum*) is a tetraploid wheat species (2*n* = 4*x* = 28, AABB) mainly grown in the Mediterranean basin, North America and the Australian wheat belt [1,2]. Compared with hexaploid wheat (bread wheat), durum wheat has higher grain protein content, strong yellow pigmentation, harder kernels, and a unique nutty flavor [3,4]. With its significant agronomic value, excellent grain quality, and versatile end use, durum wheat is considered as a staple crop in Mediterranean regions.

Grown under rain-fed conditions, durum wheat is often exposed to frequent episodes of water-deficit and heat stress [5,6]. In the field, high temperatures start to occur while soil water supply declines gradually during reproductive stages (e.g., flowering and grain filling) [4,7]. A significant number of studies have investigated the impact of the independent and combined effects of water-deficit and heat stress on plant growth, grain productivity, and grain quality in wheat [4,5,6,7,8,9]. Water-deficit and heat stress have significant impacts on photosynthetic activities, transpiration efficiency, cellular osmotic homeostasis, nutrient uptake, and metabolite production [1,4,8,10]. Such changes affect the reproductive processes in wheat, ultimately leading to changes in yield components (e.g., grain number, spikelet number, grain weight) and grain quality traits (e.g., protein content, starch content, antioxidant levels) [1,4,7,10].

Recent evidence suggests that water-deficit and heat stress could affect the stress response systems in the following generation through changes at the physiological, phenotypical and epigenetic level [11,12,13,14,15,16,17]. A few studies have demonstrated the adaptive value of the transgenerational influence of the same stress type in the offspring [13]. More interestingly, stress priming of one abiotic stress could have a beneficial impact on the occurrence of a different stress through synergistic stress signaling pathways [18,19,20,21]. For example, terminal drought stress applied in bread wheat from the reproductive stage until maturity improved the tolerance against salt stress in the next generation, mainly through changes in osmolyte accumulation, water relations modulation and lipid peroxidation [21]. As crops grown under field conditions are often exposed to multiple stressors (simultaneously, sequentially, or across multiple generations), investigation towards such phenomena (cross-stress tolerance or cross-stress effects) would be very beneficial for providing new strategies in crop breeding practices. However, it remains unknown how parental water-deficit and heat stress affect progeny performance under a different stress (e.g., nitrogen stress) in durum wheat.

Soil N availability has been a major limiting factor in wheat production. Nitrogen deficiency negatively affects the grain yield as well as grain quality in cereal crops through its impact on the nutrient uptake, photosynthesis rate, respiration efficiency, and enzyme activities [22,23,24,25,26,27,28,29]. N-stressed crop plants often have chlorotic leaves, less fertile tillers, shorter plant height and slow growth [27,28,29]. Changes in root architecture, root length, and root biomass are also known morphological responses to N starvation [28,30]. Several studies in wheat have investigated the molecular networks controlling the N stress response and N use efficiency through high-throughput approaches [28,31,32]. Specifically, a transcriptomic study in durum wheat has identified 4626 differentially expressed genes (DEGs) in response to N starvation at the grain filling stage [31]. The majority of the DEGs were nitrate or ammonium transporters, transcription factors, protein kinases and other genes involved in N assimilation. Furthermore, stress-responsive microRNAs (miRNAs) have also gained increasing attention for their essential roles in regulating plant adaptive responses to nutrient deprivation [23,25,33,34,35].

As an essential type of epigenetic regulator, miRNAs fine-tune the expression of their protein-coding target genes through post-transcriptional gene silencing [36,37,38,39,40,41,42]. In crops, miRNAs can rapidly respond to various environmental and developmental cues, playing important roles in plant growth, reproductive development and stress adaptation [36,43,44,45,46,47]. In particular, studies in durum wheat have discovered a significant number of miRNAs that play central roles in water-deficit stress, heat stress, and N-stress response networks [9,12,23,25,48,49,50,51,52,53,54]. Our previous research had shown that the parental water-deficit stress had a significant impact on the durum miRNA transcriptome in the progeny, contributing to the differences in stress response and crop performance when the next generation was exposed to water-deficit stress [11]. We have also demonstrated that the combination of parental water-deficit stress and heat stress significantly affected progeny germination traits and seedling vigor through changes in the miRNAome [12]. However, it is unknown how the miRNA-regulated N stress response networks are affected by the transgenerational effects of water-deficit and heat stress.

In this study, we characterized the morphological and physiological changes of durum wheat seedlings in response to N starvation under the transgenerational effects of water-deficit stress and heat (WH) stress in a WH stress-tolerant and WH stress-sensitive genotype. Using the small RNA sequencing approach, a systematic analysis of the miRNA expression profile subject to the progeny treatment factor, parental treatment factor, and genotype factor was performed on a genome-wide scale. To our knowledge, this is the first description of the transgenerational cross-stress effects in durum wheat. The results provide new insights to researchers and breeding programs addressing stress-tolerance improvement in cereal crops.

## 2. Results

### 2.1. Seedling Performance of Two Durum Wheat Genotypes under the Effects of Parental Water-Deficit and Heat Stress and Progeny N Starvation Stress

Two Australian durum wheat genotypes were used in this study. DBA Aurora is tolerant to water-deficit and heat (WH) stress and L6 (a University of Adelaide breeding line) is sensitive to WH stress [4]. To study the transgenerational effects of parental stress treatment, seeds of the two genotypes were collected from control (CG) and WH-stressed parents in a previous experiment [4]. There were four seed groups: AuCG (seeds from DBA Aurora parents treated with the control condition), AuWH (seeds from DBA Aurora parents treated with water-deficit and heat stress), L6CG (seeds from L6 parents treated with the control condition) and L6WH (seeds from L6 parents treated with water-deficit and heat stress). To study the response of progeny to N starvation stress, two treatment groups—control (C) and N starvation (N)—were set up for each seed group. Therefore, the current study had eight treatment groups in total: AuCG_C (DBA Aurora control parents, progeny treated with the control), AuCG_N (DBA Aurora control parents, progeny treated with N starvation), AuWH_C (DBA Aurora water-deficit and heat stress parents, progeny treated with the control), AuWH_N (DBA Aurora water-deficit and heat stress parents, progeny treated with N starvation), L6CG_C (L6 control parents, progeny treated with the control), L6CG_N (L6 control parents, progeny treated with N starvation), L6WH_C (L6 water-deficit and heat stress parents, progeny treated with the control), and L6WH_N (L6 water-deficit and heat stress parents, progeny treated with N starvation).

To evaluate seedling performance, eight morphological and physiological traits were measured at the three-week stage: seedling height, leaf number, shoot fresh weight, shoot dry weight, root fresh weight, root dry weight, primary root length, and chlorophyll content (Table 1 and Table 2). The growth and development of durum wheat seedlings were significantly affected by N deficiency. For progeny groups with the same parental origin, all eight traits showed a significant reduction under N starvation stress when compared with their control (i.e., AuCG_N vs. AuCG_C, AuWH_N vs. AuWH_C, L6CG_N vs. L6CG_C, and L6WH_N vs. L6WH_C). However, the % reduction of each trait varied across progeny groups in the two genotypes. For example, for plant height, progeny from the WH parents appeared to have a lower percentage reduction in response to N starvation when compared with the progeny from the control parents in both genotypes. In DBA Aurora, the percentage reduction of plant height was 32.8% between AuWH_N vs. AuWH_C, while the percentage reduction was 34.5% between AuCG_N vs. AuCG_C (Table 1). In L6, the percentage reduction of plant height was 34.3% between L6WH_N vs. L6WH_C, while the percentage reduction was 35.9% between L6CG_N vs. 6CG_C (Table 2). A similar pattern was also observed for the primary root length (11.3% (AuWH_N vs. AuWH_C) and 13.8% (AuCG_N vs. AuCG_C) in DBA Aurora; 13.8% (L6WH_N vs. L6WH_C) and 14.2% (L6CG_N vs. 6CG_C) in L6).

For the other six traits (leaf number, shoot fresh weight, shoot dry weight, root fresh weight, root dry weight, primary root length, and chlorophyll content), a genotype-dependent pattern can be observed when it comes to the transgenerational effects of parental treatment. For DBA Aurora, the WH stress-tolerant variety, the parental exposure of WH helped to mitigate the negative effects of N starvation in the progeny (lower percentage reduction for the traits measured). For example, in DBA Aurora, the percentage reduction of chlorophyll content in progeny groups from the AuCG parents was 20.6%, while in the progeny from the AuWH parents the percentage reduction was 18.1% (Table 1). In contrast, for the WH-sensitive genotype L6, the parental exposure of WH exacerbated the negative impacts of N starvation. For example, for shoot fresh weight, the percentage reduction in progeny from the L6CG parents was 63.9%, while in progeny groups from the L6WH parents, the reduction rate was 65.6% (Table 2).

### 2.2. Durum Wheat MiRNA Expression Profile across Different Treatment Groups

To investigate how miRNAs are involved in the N starvation response under the effects of transgenerational WH stress, eight sRNA libraries were constructed and sequenced from the eight treatment groups (AuCG_C, AuCG_N, AuWH_C, AuWH_N, L6CG_C, L6CG_N, L6WH_C, L6WH_N). In total, over 143 million raw reads were generated with over 51 million reads being unique (Appendix A). After filtering and data processing, a total of 102.05 million clear sRNA reads were obtained, of which over 45 million were unique sRNA reads (Appendix A). Through the bioinformatics pipeline, a total of 1534 miRNA-MIR entries (different combinations of mature miRNA products and MIR origins in the genome) were identified, with 190 being novel miRNAs (Appendix A). The identified miRNAs were grouped into five categories (group 1 to 5). The definition and selection criteria for each group were described previously [12]. Briefly, groups 1 to 4 contained different types of conserved miRNAs, and group 5 included all the novel miRNAs identified in this study. The conserved miRNAs belonged to 61 MIR families (Appendix A). Group 2 (definition: sRNA reads can be mapped to miRNA references in the miRBase, but the pre-miRNA sequence cannot be mapped to the durum wheat genome; however, the sRNA reads can be mapped to the genome while the extended genome sequences from the mapped location can form secondary hairpins) contains the highest number of miRNA entries (895). Group 4 (definition: sRNA reads can be mapped to miRNA references in the miRbase; however, either the pre-miRNA or sRNA reads can be mapped to the durum wheat genome) contains the lowest number of miRNA entries (41). The miRNA conservation profile across different reference plant species in miRBase are shown in Figure 1a. The identified miRNAs in the current study showed the highest degree of conservation with *Triticum aestivum* as expected, and the lowest degree of conservation with *Triticum turgidum*, due to the limited number of ttu-miRNA entries registered in the miRBase.

Among all the miRNA entries, 174 miRNAs were considered as highly expressed (higher than the dataset average) based on their normalized reads count (Appendix A). The expression of 819 miRNAs were considered as medium, of which the normalized reads count was lower than the dataset average but was higher than 10 (Appendix A). A total of 541 miRNAs were lowly expressed, of which the normalized reads count was lower than 10. The distribution of the identified miRNAs across different treatment groups are shown in Figure 1b. In DBA Aurora, 569 miRNAs were commonly expressed across the four treatment groups AuCG_C, AuCG_N, AuWH_C, and AuWH_N (Figure 1b). In L6, 681 miRNAs were commonly expressed across L6CG_C, L6CG_N, L6WH_C, and L6WH_N (Figure 1b). Interestingly, in DBA Aurora, the control progeny from the control parents (AuCG_C) had the highest number of exclusively expressed miRNAs (140). In L6, the N starvation progeny from the stressed parents (L6WH_N) had the highest number of exclusively expressed miRNAs (136).

### 2.3. Differentially Expressed MiRNAs (DEMs) Subject to Different Factors

The expression analysis of differentially expressed miRNAs (DEMs) was performed based on the miRNA normalized reads count across different treatment groups. To investigate the effects of progeny N starvation on miRNA expression, pairwise comparisons were made between control and N starvation treatment groups with the same parental origin (Appendix A). Between AuCG_N and AuCG_C, 672 miRNAs showed significant differential expression in response to N starvation; between AuWH_N and AuWH_C, 477 miRNAs showed significant differential expression; between L6CG_N and L6CG_C, 542 miRNAs showed significant differential expression; between L6WH_N and L6WH_C, 494 miRNAs showed significant differential expression (Appendix A). Interestingly, a consistent pattern can be found regarding the number of significantly down-regulated and up-regulated miRNAs. In both genotypes, the number of down-regulated miRNAs was always higher than the number of up-regulated miRNAs within each pairwise comparison of N starvation responsive miRNAs (Figure 2a). Given the gene-silencing effects of miRNAs, the pattern suggests that the N starvation-responsive miRNAs were more involved in promoting the expression of their target genes rather than repressing gene expression.

To investigate the effects of parental treatment on progeny miRNA expression, pairwise comparisons were made between groups with the same progeny treatment factor but were originated from different parental sources (Appendix A). Between AuWH_C and AuCG_C, 319 miRNAs showed significant differential expression due to parental WH stress; between AuWH_N and AuCG_N, 471 miRNAs showed significant differential expression; between L6WH_C and L6CG_C, 469 miRNAs showed significant differential expression; between L6WH_N and L6CG_N, 342 miRNAs showed significant differential expression (Appendix A). The number of significantly down-regulated and up-regulated miRNAs in each pairwise comparison was also compared (Figure 2). Except for AuWH_C vs. AuCG_C, the number of up-regulated miRNAs was always higher than the number of down-regulated miRNAs (Figure 2b).

To identify miRNAs with genotype-dependent expression patterns, pairwise comparisons were made between groups with the same progeny treatment and parental treatment of two different genotypes (Appendix A). Between AuCG_C and L6CG_C, 363 miRNAs showed significant genotypic expression; between AuCG_N and L6CG_N, 620 miRNAs showed significant genotypic expression; between AuWH_C and L6WH_C, 601 miRNAs showed significant genotypic expression; between AuWH_N and L6WH_N, 232 miRNAs showed significant genotypic expression (Appendix A). Regardless of the parental and progeny treatment conditions, the number of down-regulated miRNAs (i.e., miRNA was less abundant in DBA Aurora) was always higher than the number of up-regulated miRNAs (i.e., miRNA was more abundant in DBA Aurora) (Figure 2c). As lower miRNA expression would allow for higher expression of their target genes, the pattern suggests more miRNAs were promoting the expression of their target genes in DBA Aurora.

### 2.4. Analyses of the Functional Target Genes of DEMs

In the current study, protein-coding genes targeted by DEMs in the two durum wheat genotypes were identified using the GSTAr package. The recently published durum wheat assembly Svevo.v1 was used as the reference. Gene ontology (GO) analysis was performed as previously described [11,12,53,54] to provide more functional information on the mRNA targets regarding the three GO categories: cellular component (CC), biological process (BP), and molecular function (MF). KEGG (Kyoto Encyclopedia of Genes and Genomes) pathway analysis (http://www.genome.jp/kegg, accessed on 14 January 2021) was conducted [11,12,53,54] to determine the biological pathways in which miRNA-target genes participate.

First, gene transcripts targeted by N starvation-responsive DEMs were investigated (Appendix A). For the 672 miRNAs differentially expressed between AuCG_N and AuCG_C, 15,794 target transcripts were identified (Appendix A). For the 477 miRNAs differentially expressed between AuWH_N and AuWH_C, 11,787 target transcripts were identified (Appendix A). For the 542 miRNAs differentially expressed between L6CG_N and L6CG_C, 12,102 target transcripts were identified (Appendix A). For the 494 miRNAs differentially expressed between L6WH_N and L6WH_C, 15,202 target transcripts were identified (Appendix A). GO enrichment analysis showed the top GO terms enriched for target transcripts identified in each pairwise comparison subject to the N starvation factor (Figure 3). Certain GO terms were common among treatment groups while others were specific to the genotype or parent treatment factor. For example, the term protein binding (GO:0005515, MF category) was enriched for all comparisons (except for AuCG_N vs. AuCG_C). Cell differentiation (GO:0030154, BP category) was only enriched for the targets in DBA Aurora comparison groups but not in L6. Although there were no L6-specific GO terms enriched for both L6 comparisons, there were GO terms that were specifically enriched for individual L6 comparisons. For example, the cell surface receptor signaling pathway (GO:0007166, BP category) was specifically enriched for transcripts targeted by progeny N-responsive miRNAs in L6 groups from the stressed parents (L6WH_N vs. L6WH_C), and xylem development (GO:0010089, BP category) was specifically enriched for L6CG_N vs. L6CG_C. The top five significant KEGG pathways enriched for each comparison are listed in Table 3. The results provide further information on the specific functions of the target genes of N starvation-responsive miRNAs. It is noted that the ko04010–MAPK signaling pathway was ranked as the top KEGG pathway for all the N starvation factor comparisons except for L6CG_N vs. L6CG. In general, most of the top enriched pathways were related to signaling (e.g., ko04070 and ko00562 that mediate Ca^2+^ signaling), protein processing (e.g., ko04141) and N metabolism (e.g., ko00230).

Secondly, gene transcripts targeted by DEMs responsive to the transgenerational effects of parental treatment were identified (Appendix A). For the 319 miRNAs differentially expressed between AuWH_C and AuCG_C, 7273 target transcripts were identified (Appendix A). For the 471 miRNAs differentially expressed between AuWH_N and AuCG_N, 9671 target transcripts were identified (Appendix A). For the 469 miRNAs differentially expressed between L6WH_C and L6CG_C, 9946 target transcripts were identified (Appendix A). For the 342 miRNAs differentially expressed between L6WH_N and L6CG_N, 9115 target transcripts were identified (Appendix A). GO enrichment analysis identified the top GO terms enriched for target transcripts of DEMs in each pairwise comparison (Figure 4). Similarly, certain GO terms were common among the comparison groups, while others were specific to the genotype or progeny treatment factor. For example, RNA interference (GO:0016246, BP category) was only enriched for comparison groups in DBA Aurora (AuWH_C vs. AuCG_C, AuWH_N vs. AuCG_N). Xylem and phloem pattern formation (GO:0010051, BP category) was enriched for comparison groups where progeny was treated with N starvation (AuWH_N vs. AuCG_N and L6WH_N vs. L6CG_N). The top five significant KEGG pathways enriched for each comparison are listed in Table 4. The ko04010–MAPK signaling pathway ranked as the top pathway again for two parental treatment factor comparisons (AuWH_N vs. AuCG_N and L6WH_C vs. L6CG_C). It was also observed that ko03013–RNA transport was only enriched for AuWH_N vs. AuCG_N, and ko03008–Ribosome biogenesis in eukaryotes was only enriched for L6WH_N vs. L6CG_N.

Finally, gene transcripts targeted by DEMs with a genotype-dependent pattern were analyzed (Appendix A). For the 363 miRNAs differentially expressed between AuCG_C and L6CG_C, 7772 target transcripts were identified (Appendix A). For the 620 miRNAs differentially expressed between AuCG_N and L6CG_N, 15,762 target transcripts were identified (Appendix A). For the 601 miRNAs differentially expressed between AuWH_C and L6WH_C, 13,177 target transcripts were identified (Appendix A). For the 232 miRNAs differentially expressed between AuWH_N and L6WH_N, 4957 target transcripts were identified (Appendix A). There seems to be a higher level of specificity of the GO terms enriched across these pairwise comparisons compared to previous comparisons (Figure 5). Many terms were specific to the parental or progeny treatment factor. For example, RNA interference (GO:0016246, BP category) and the cellular response to unfolded protein (GO: 0034620, BP category) were only enriched for AuCG_C vs. L6CG_C; the miRNA catabolic process (GO:0010587, BP category) and auxin-activated signaling pathway (GO:0009734, BP category) were only enriched for AuCG_N vs. L6CG_N (Figure 5). Plant-type cell wall biogenesis (GO:0009832, BP category) and protein serine/threonine kinase activity (GO:0004674, MF category) were only enriched for AuWH_C vs. L6WH_C; the methionine biosynthetic process (GO:0009086, BP category) and aspartate kinase activity (GO:0004072, MF category) were only enriched for AuWH_N vs. L6WH_N (Figure 5). The results provide more insights on the specific biological functions and molecular pathways with which genotype-dependent miRNA-target genes were associated. The top five significant KEGG pathways enriched for each comparison are listed in Table 5. The MAPK signaling pathway was common to three genotype factor comparisons (AuCG_C vs. L6CG_C, AuCG_N vs. L6CG_N and AuWH_C vs. L6WH_C). The mRNA surveillance pathway (ko03015) was only enriched for AuCG_C vs. L6CG_C; the RNA degradation pathway (ko03018) was only enriched for AuCG_N vs. L6CG_N; the Toll-like receptor signaling pathway (ko04620), Toll and Imd signaling pathway (ko04624), and NOD-like receptor signaling pathway (ko04621) were only enriched for AuWH_C vs. L6WH_C; RNA polymerase (ko03020), cysteine, and methionine metabolism (ko00270) and the ABC transporters pathway (ko02010) were only enriched for AuWH_N vs. L6WH_N (Table 5).

### 2.5. QPCR Analysis of DEMs and Their Target Genes

qPCR expression analysis was performed on six N stress-responsive miRNAs (ttu-miR160, ata-miR167b-3p, ata-MIR169d-p3, osa-miR393a_L+1R+2, tae-miR398_L-1R+1 and osa-miR827) and three target genes (two leucine-rich repeat receptor-like protein kinase family protein genes (*LRR-PK*) targeted by ata-MIR169d-p3, and a pentatricopeptide repeat-containing protein gene (*PPR*) targeted by osa-miR393a_L+1R+2) to validate their expression (Figure 6). Most miRNAs showed a similar expression pattern to their sRNA sequencing profile. Two miRNAs (ata-MIR169d-p3 and osa-miR393a_L+1R+2) exhibited a significant down-regulation pattern in response to progeny N starvation treatment, irrespective of parental origin and genotype. Their target genes exhibited a significant up-regulation pattern in response to progeny N starvation treatment, which was expected given that lower miRNA expression would allow for higher expression of their targets. For the *LRR-RK* genes (TRITD2Av1G130230 and TRITD2Bv1G123760), the highest up-regulation fold-change was found in the L6 progeny from the WH parents (L6WH_N vs. L6WH_C). However, for the *PPR* gene, the highest up-regulation fold change was found in the DBA Aurora progeny from the CG parents (AuCG_N vs. AuCG_C).

## 3. Discussion

Crop plants growing under field conditions are often challenged by more than one abiotic or biotic constraint, simultaneously, sequentially, or even across generations. Plants have evolved sophisticated response systems to cope with stress by activating a range of changes at the cellular, molecular, physiological, and phenotypic levels [36,55,56,57]. Understanding the response mechanisms that underpin stress tolerance level is critical for the development of next-generation crop varieties with higher environmental resilience. Recently, the phenomenon of transgenerational stress memory in plants has become of increasing research interest [11,12,13,14,17,21,58,59]. Crop varieties could exhibit genotype-dependent responses to reoccurring stress or a second stress factor under the transgenerational effects of parental stress [11,12,13,60].

As a key nutrient required for crop development, N is one of the most limiting nutrients for yield production. The lack of N often leads to a serious delay in development and significant symptoms [28,29,30,61]. In the current study, the WH-stress tolerant genotype DBA Aurora and WH stress-sensitive genotype L6 both exhibited a significant reduction in the morphological and physiological traits measured, including seedling height, leaf number, shoot fresh weight, shoot dry weight, root fresh weight, root dry weight, primary root length, and chlorophyll content (Table 1 and Table 2). These results are consistent with previous studies except for changes in root-related traits, where observations of both N stress-induced and N stress-inhibited root traits have been recorded. In a recent study using a Chinese bread wheat cultivar (Wanmai 52), low nitrogen stress significantly inhibited both seedling growth and root development [28]. Growth parameters including plant height, leaf area, shoot dry weight, root dry weight, total root length, and total root number all showed a significant reduction after 10 days of low N stress treatment. For another study using a different Chinese bread wheat cultivar (Yumai 34), different response patterns of morphological traits were recorded [29]. At the eight-day seedling stage, plant height, shoot fresh weight, and shoot dry weight showed a significant decrease under N starvation, but root length, root fresh weight, and root dry weight all showed a significant increase in response to stress. For a study using two highly N-responsive bread wheat genotypes (Kalyansona and NP-890) grown hydroponically, genotype-dependent trends were observed for the morphological traits measured [30]. Both genotypes exhibited a significant reduction in shoot length, shoot fresh weight, and shoot dry weight under N starvation. However, Kalyansona showed reduced root fresh weight and root dry weight in response to N stress, while NP-890 exhibited no significant difference in these two traits between N-optimum and N-starved conditions. Furthermore, in a study using two Italian durum wheat cultivars (Ciccio and Svevo), after a month of chronic N stress applied within a hydroponic system, no significant differences were found for plant height and leaf area between stressed and non-stressed Z14 stage seedlings [25]. However, Ciccio N-stressed plants had significantly longer root length (doubled) whereas the difference in Svevo was much smaller. The differences in the morphological responses of N stress (particularly root-related traits) across these studies could be due to the genetic variability of the cultivars used, different growing systems (hydroponics or soil-based), and the time of stress application and measurement (different seedling developmental stages).

Interestingly, in the current study, although both DBA Aurora and L6 showed a significant reduction in all the morphological and physiological traits, it was noted that the genotype and parental stress factors also had an influence on the reduction rate of certain traits under N stress. In DBA Aurora, the WH stress-tolerant variety, parental stress treatment helped to lower the reduction rate of all traits. However, in L6, the WH-stress sensitive genotype, most of the traits had a higher percentage reduction in the progeny from the stressed parents when compared with the progeny from the control parents, except for two traits (seedling height and primary root length). A few studies have recorded a similar pattern, where the transgenerational effects of stress showed genotype-dependent differences in the progeny. In field-grown peanut, water-saving irrigation practices in the parents had positive impacts in the early seedling growth in the next generation, but how transgenerational stress memory was expressed was dependent on the genotype and the offspring’s environmental conditions [13]. In rice, six consecutive generations of drought stress had a cumulative effect on the DNA methylation pattern of two rice cultivars with contrasting drought tolerance levels, in which the stress-resistant cultivar had a higher percentage of stably transmitted methylated loci [60]. In another study in rice, four rice genotypes with contrasting tolerance levels to salinity stress also exhibited genotype-dependent transgenerational alteration in DNA methylation under both salinity and alkaline stresses [62]. Moreover, our previous study in Australian durum wheat demonstrated that the transgenerational influence of water-deficit stress on the physiological traits, yield components, and grain quality traits in the next generation varied depending on the genotype [11]. Therefore, it can be concluded that the impacts of transgenerational stress are genotype-dependent, even within the same crop species, and should not be generalized. Future research focusing on the mechanisms of transgenerational stress memory should aim to include a wide range of germplasm containing varieties with contrasting levels of stress tolerance.

Most of the previous studies evaluating transgenerational stress effects have focused on the same stress type across generations. Our study represents the first investigation of transgenerational cross-stress effects in durum wheat. Cross-stress tolerance refers to the improved ability of plants to tolerate different types of stress after the exposure of a primary stress type [20,63]. Achieving cross-stress tolerance in plants will likely require fine-tuned coordination of synergistic or antagonistic stress signaling pathways shared across different stress response systems. Key participants that have received a lot of attention in the crosstalk of stress response include plant hormones, transcription factor (TF) families, reactive oxygen species (ROS), reactive nitrogen species (RNS) and miRNAs. Plant miRNAs modulate a wide range of biological processes including stress signal recognition, hormone signal transduction, nutrient metabolism, transport, and cellular homeostasis [36,41,44,45]. For example, the metabolism and signaling of almost all phytohormones (such as auxin, abscisic acid (ABA), gibberellic acid (GA), jasmonic acid (JA), brassinosteroid, ethylene, and cytokinin) are regulated by different miRNA families [64,65,66,67]. A significant portion of the target repertoire of miRNAs are TF families, including DREB, NAC, ARF, WRKY, MYB, MYC, SPL, bZIP, GRF, NF-YC, ERF, bHLH, and GATA family members [53,54,68,69]. Moreover, miRNA families such as miR398 and miR528 are hub regulators that control cellular homeostasis through targeting redox-related enzymes such as superoxide dismutases (SODs), peroxidases (PODs) and ascorbate oxidase (AAOs) [70,71,72]. With the small RNA high-throughput sequencing technology and advanced bioinformatics pipelines, many studies have investigated the durum wheat miRNA expression profile under different stress conditions and at various developmental stages [9,23,25,50,52,73], including our previous research with a particular focus on water-deficit and heat stress [11,48,49,51,53,54]. Given their roles in stress-regulatory crosstalk, miRNAs would also be expected to play a significant part in transgenerational cross-stress responses in plants.

In the current study, we provided the first systematic and detailed analysis of the durum wheat miRNA population subject to the progeny N starvation, parental WH stress and genotype factors. Expression profiles of 1534 miRNAs (including 190 novel miRNAs) were provided with significant DEMs identified in the two durum wheat genotypes (Figure 2). The highest number of N starvation-responsive DEMs was found in DBA Aurora progeny groups from the control parents (AuCG_N vs. AuCG_C). The highest number of DEMs that were responsive to the parental stress treatment factor was found in DBA Aurora progeny groups treated with N starvation (AuWH_N and AuCG_N). The results suggest that DBA Aurora was more responsive to N starvation stress than L6, and the expression of N stress-responsive miRNAs in DBA Aurora was influenced by parental treatment. Given that changes in miRNA expression directly affect the expression of functional target genes, GO and KEGG enrichment analyses of the target genes provide a collective view of the key biological processes regulated by the stress-responsive miRNA-target modules. For example, one of the most-represented KEGG pathways of the targets of different types of DEMs was the ko04010–MAPK signaling pathway (Table 3, Table 4 and Table 5). MAPK signaling cascades are widely involved in plant developmental stages and in response to different types of abiotic and biotic stresses [74,75]. In the current study, a major target gene family involved in the MAPK signaling pathway were leucine-rich repeat receptor-like protein kinase family protein genes, targeted by the miR169 family (e.g., TRITD2Bv1G123760 and TRITD2Av1G130230 targeted by ata-MIR169d-p3, Appendix A). Notably, ata-MIR169d-p3 was significantly down-regulated under N starvation stress in both genotypes and parental origins: −7.63 fold for AuCG_N vs. AuCG_C, −6.47 fold for AuWH_N vs. AuWH_C, −5.36 fold for L6CG_N vs. L6CG_C and −5.47 fold for L6WH_N vs. L6WH_C, respectively (Appendix A). Reduced expression of ata-MIR169d-p3 would promote the expression of its target genes. Indeed, this regulatory pattern was validated via qPCR (Figure 6). In durum wheat, transcriptome-sequencing has shown that many leucine-rich repeat receptor proteins increased expression in response to nitrogen starvation [31]. A study in Arabidopsis demonstrated that leucine-rich repeat receptor kinases (LRR-RKs) are critical receptors of root-derived CEPs (C-terminally encoded peptides) that mediate root-to-shoot N-demand signaling under N starvation [76]. The up-regulation of leucine-rich repeat receptor genes via lowered miR169 abundance likely contributes to the adaptation of durum wheat to N stress via promoted N stress signaling. Future research can further investigate the regulatory pattern of the miR169-LRR-RKs in a wider range of germplasm under different levels of N stress, together with the measurement of mobile CEPs involved in root-to-shoot signaling.

It is also important to investigate GO and KEGG pathway terms that were exclusively enriched for certain DEM comparison groups. For example, when looking at the targets of N starvation-responsive DEMs (Figure 3), the GO term cell differentiation was only enriched for DBA Aurora. The process of nitrogen assimilation is essential for cell differentiation and biomass production during crop growth [77]. The results suggest that miRNA target genes that participate in cell growth and differentiation were more responsive to N stress in DBA Aurora, possibly contributing to the lower biomass percentage reduction in DBA Aurora when compared with L6. As another example, when looking at genes targeted by DEMs that were responsive to parental treatment factor (Figure 4), the GO term xylem and phloem pattern formation was only enriched for N-stressed comparison groups (AuWH_N vs. AuCG_N and L6WH_N vs. L6CG_N). Moreover, when looking at genes targeted by genotype-dependent DEMs, the KEGG pathway ABC transporters (ko02010) were only enriched for N-stressed comparison groups from the stressed parents (AuWH_N vs. L6WH_N). In vascular plants, the long-distance transport of water, nutrients and phytohormones rely on xylem and phloem structure and function [78,79]. Both water deficiency and heat stress challenge plant water uptake and the vascular transport system and have pronounced effects on crop reproduction through physiological signals such as xylem-borne ABA [55,80,81,82]. The nitrate loading and unloading processes between xylem and phloem are also critical for nitrate redistribution to optimize plant growth, with the interplay between ABA transporters and nitrate transporters [83,84,85,86]. It is possible that the parental WH treatment would have transgenerational effects on xylem and phloem function via miRNA-regulated genes associated with long-distance transport, such as xylem- or phloem-located nitrate transporters and ABA transporters [84,85]. The key in utilizing transgenerational cross-stress effects in breeding relies on identifying miRNA and target gene candidates that are both associated with the WH stress response and N stress tolerance in different genotypes.

To achieve this goal, we looked into the regulatory patterns of individual N stress-responsive miRNA candidates, to identify those that were either specific to the stress tolerant genotype or parental stress treatment. As an example, ata-miR164c-3p was only significantly down-regulated under N starvation in DBA Aurora: −1.61 fold for AuCG_N vs. AuCG_C and −1.18 fold for AuWH_N vs. AuWH_C, respectively (Appendix A), while this miRNA did not exhibit any expression difference in L6 in response to N starvation-stress. ata-miR164c-3p has 113 identified target transcripts (Appendix A), among which there are two high-affinity nitrate transporter genes NRT2.1 (TRITD6Av1G006030 and TRITD6Bv1G008690). Plants mainly uptake nitrogen from the soil through nitrate (NO_3_^−^) and ammonium (NH_4_^+^) transporters [87]. The two types of NO_3_^−^ transport systems, low-affinity and high-affinity transporters are in charge of nitrate absorption and remobilization. The low-affinity transport system (LATS) mainly functions under high soil N conditions, whereas the high-affinity transport system (HATS) is responsible for scavenging NO_3_^−^ ions under low soil N conditions [88]. The increased activities of high-affinity nitrate transporters are well-documented adaptive responses to enhance the efficiency of nitrogen uptake. Studies in Arabidopsis have shown that NRT2.1 has central functions in regulating root development to maximize external NO_3_^−^ availability and interacting with the ethylene biosynthesis and signaling pathway to fine-tune nitrate acquisition in a feedback loop [89,90]. The lowered expression of ata-miR164c-3p would allow for the higher expression of its NRT2.1 targets, contributing to the adaptation of N stress. The reduction of ata-miR164c-3p under nitrogen stress has also been observed in a previous study in Italian durum wheat cultivars, where ata-miR164c-3p (ttu-miR164b) showed lower expression in the leaf tissue of both cv. Ciccio and Svevo and the root tissue of Svevo in response to chronic nitrogen (N) stress [25]. Here, the positive promotion of NRT2.1 via repressed miR164 expression was genotype-dependent (specific to DBA Aurora), possibly due to the pedigree differences in the germplasm. It is also worth noting that ata-miR164c-3p did not exhibit any significant expression change subject to the parental treatment factor, indicating that miR164-NRT2.1 could be a consistent N regulatory module not affected by transgenerational stress.

The regulatory functions of many durum wheat miRNAs, however, were affected by the parental stress treatment. For example, for the DBA Aurora N-stressed progeny groups, osa-miR393a_L+1R+2 had a significantly lower expression (−3.44 fold) in the progeny from the WH-stressed parents when compared with the progeny from the control parents (AuWH_N vs. AuCG_N) (Appendix A). qPCR has also shown that osa-miR393a_L+1R+2 was significantly down-regulated in response to N starvation in both genotypes (Figure 6). One of the targets of osa-miR393a_L+1R+2 is a pentatricopeptide repeat (*PPR*) protein gene (Appendix A). *PPR* proteins are one of the biggest groups of protein families in land plants [91]. Over the last decade, research has shown the regulatory functions of *PPR* proteins in cereal crops [92,93,94,95]. The majority of the *PPR* proteins have been located in mitochondria and chloroplasts, and they universally function in regulating organellar gene expression through sequence-specific recognition of RNA sequences and participation in processes such as RNA splicing, RNA editing and RNA translation [91,92,93,94,95]. In rice, a *PPR* protein localized in the chloroplast is required for proper chloroplast development and seedling growth under cold stress for its important functions in maintaining photosynthetic electron transport [95]. In maize, a *PPR* gene located within a kernel size-related quantitative trait locus *qKW9* is essential for optimizing the photosynthesis rate during grain filling through its function in editing a subunit of the chloroplast NADH dehydrogenase-like complex [92]. In the current study, lowered osa-miR393a_L+1R+2 expression in response to N starvation allowed for a higher expression of the *PPR* gene (Figure 6), which could serve a similar positive function in optimizing chloroplast development and the photosynthetic rate in durum wheat seedlings under nitrogen stress. Future research could look into the spatial–temporal expression of osa-miR393a_L+1R+2 and *PPR* proteins in various cellular compartments in the leaves under nitrogen stress, to further affirm the function of the osa-miR393a_L+1R+2-PRR module in chloroplast development and photosynthesis in durum wheat.

In conclusion, the current research characterized the morphological, physiological, and molecular responses of Australian durum wheat seedlings to N starvation under the influence of parental water-deficit and heat stress. We provide the first description of the durum wheat miRNAome with specific expression patterns subject to the progeny N stress, parental treatment, and genotype factors, along with the annotation of their functional target genes. The results suggest that the cross-stress impacts of transgenerational stress are genotype-dependent in durum wheat. Novel insights gained on the molecular level further confirmed the importance of miRNAs in the stress adaptation processes of durum wheat and have facilitated new opportunities for engineering cross-tolerance across generations in crop plants.

## 4. Materials and Methods

### 4.1. Plant Growing Conditions

Two Australian durum wheat genotypes were used in this study. DBA Aurora is tolerant to WH stress and L6 is sensitive to WH stress. Four seed groups were collected from a previous glasshouse experiment: AuCG (seeds from DBA Aurora parents treated with control condition), AuWH (seeds from DBA Aurora parents treated with water-deficit and heat stress), L6CG (seeds from L6 parents treated with control condition), and L6WH (seeds from L6 parents treated with water-deficit and heat stress). Two treatment groups—control (C) and N starvation (N)—were set up for each seed group. There were eight treatment groups in total: AuCG_C (DBA Aurora control parents, progeny treated with the control), AuCG_N (DBA Aurora control parents, progeny treated with N starvation), AuWH_C (DBA Aurora water-deficit and heat stress parents, progeny treated with the control), AuWH_N (DBA Aurora water-deficit and heat stress parents, progeny treated with N starvation), L6CG_C (L6 control parents, progeny treated with the control), L6CG_N (L6 control parents, progeny treated with N starvation), L6WH_C (L6 water-deficit and heat stress parents, progeny treated with the control), and L6WH_N (L6 water-deficit and heat stress parents, progeny treated with N starvation).

Young seedlings were grown in a controlled glasshouse environment (24/18 °C, 16/8 h photoperiod) as previously described [12]. Each treatment group had 12 individual biological replicates (one plant per pot). Each pot contained 1.2 kg of N40 sand with 0.5% CaCO_3_ [1,4]. Complete basal nutrient solution [1,4] was provided to progeny control treatment groups (AuCG_C, AuWH_C, L6CG_C, L6WH_C). The progeny N starvation treatment groups (AuCG_N, AuWH_N, L6CG_N, L6WH_N) were supplied with the basal nutrient solution without the nitrogen component.

### 4.2. Seedling Measurement and Statistical Analysis

At three weeks, six seedlings per treatment group were sampled for molecular experiments. Fresh seedling shoots of the six biological replicates were taken and snap-frozen in liquid nitrogen for downstream experiments. The remaining six biological replicates per group were used to evaluate seedling growth performance. Chlorophyll content was measured on the youngest fully expanded leaf of each seedling using a SPAD meter [1,4]. The seedling height (cm), leaf number, shoot fresh weight (g), root fresh weight (g), and primary root length (cm) were recorded. The shoot dry weight (g) and root dry weight (g) were determined after overnight drying of the fresh shoot and root at 65 °C. Two-way-ANOVA analysis was performed to determine the statistical significance of each trait at *p* < 0.05 under the impact of parental WH stress × progeny N starvation using Genstat (20th Edition, VSN International). Results were shown as the mean ± SE (*n* = 6).

### 4.3. Small RNA Sequencing Analysis of Conserved and Novel Durum MiRNAs

Total RNA was extracted from frozen seedling shoot material using the Tri reagent as previously described (Sigma-Aldrich, North Ryde, Australia) [11,53,54]. Extracted total RNA samples were treated with TURBO DNase (ThermoFisher Scientific, Scoresby, Australia) following the manufacturer’s instructions. The concentration, quantity and quality of purified total RNA samples were measured on a NanoDrop Lite spectrophotometer (ThermoFisher Scientific, Scoresby, Australia). RNA integrity was assessed by agarose gel electrophoresis and Bioanalyzer measurement. High-quality RNA samples from the six biological replicates per treatment group were equally pooled for small RNA sequencing purpose. A total of eight small RNA libraries (AuCG_C, AuCG_N, AuWH_C, AuWH_N, L6CG_C, L6CG_N, L6WH_C, L6WH_N) were made using the TruSeq Small RNA Sample Prep Kit. Small RNA sequencing (single-end, 50 bp) was performed on an Illumina HiSeq2500 at LC-Bio (Hangzhou, China). The sequencing datasets were deposited in the NCBI GEO database (accession number GSE168094). The bioinformatics pipeline for identifying conserved and novel durum wheat miRNAs was employed as previously described [11,12,53,54]. Briefly, miRNA annotation was performed with the in-house ACGT101-miR program (LC Sciences, Houston, TX, USA). Conserved miRNAs were identified via BLAST search against reference miRNA entries in the public plant miRNA repository miRBase (Release 22.1). Mapped miRNA reads were aligned to the durum wheat reference genome (Svevo.v1) to determine their genomic location. Unmapped reads were used to identify novel durum miRNAs using the RNAfold (http://rna.tbi.univie.ac.at/cgi-bin/RNAfold.cgi, accessed on 15 December 2020) tool based on their secondary miRNA hairpin structures. The identified miRNAs were categorized into five groups (group 1 to 5). Groups 1 to 4 contained different categories of conserved miRNAs. Group 5 consisted of novel durum miRNA identified in this study.

Read normalization across the sequenced libraries was carried out as previously described to enable miRNA expression analysis [11,12,53,54]. Differentially expressed miRNAs (DEMs) were identified based on the normalized count of each miRNA across the eight sequenced libraries. The chi-square 2 × 2 test and Fisher’s test were used for determining the statistical significance in each pairwise comparison. DEMs with *p*-value < 0.05 and|log2 (Fold change)| > 1 in each comparison were identified.

### 4.4. Identification of MiRNA Target Genes and Functional Annotation

In silico target analysis was performed using the GSTAr software to identify protein-coding gene transcripts targeted by durum DEMs at the post-transcriptional level. Gene ontology (GO) analysis was performed as previously described [11,12,53,54] to provide more functional information on the mRNA targets regarding the three GO categories: the cellular component (CC), biological process (BP), and molecular function (MF). KEGG (Kyoto Encyclopedia of Genes and Genomes) pathway analysis (http://www.genome.jp/kegg, accessed 14 January 2021) was conducted [11,12,53,54] to determine the biological pathways that miRNA-target genes participate in.

### 4.5. QPCR Analysis of Selected MiRNAs and Target Genes

Six N stress-responsive miRNAs (ttu-miR160, ata-miR167b-3p, ata-MIR169d-p3, osa-miR393a_L+1R+2, tae-miR398_L-1R+1 and osa-miR827) and three target genes (two leucine-rich repeat receptor-like protein kinase family protein genes (*LRR-PK*) targeted by ata-MIR169d-p3, and a pentatricopeptide repeat-containing protein gene (*PPR*) targeted by osa-miR393a_L+1R+2) were selected for qPCR expression analysis. cDNA synthesis was performed with DNase-treated RNA samples using the MystiCq microRNA cDNA Synthesis Mix (Sigma-Aldrich) as described previously [11,53,54]. qPCR was performed on an Applied Biosystems ViiA7 Real-Time PCR machine using the PowerUp SYBR Green Master Mix [11,53,54]. The relative gene expression was calculated using the *Triticum turgidum* GAPDH as the reference gene. Results were presented as log2 (fold-change) based on the relative gene expression of three biological replicates. Student’s *t*-test was used to detect statistical significance at *p* < 0.05 for each pairwise comparison. Details of the primers used in this study are listed in Appendix A.

## Figures and Tables

**Figure 1 plants-10-00826-f001:**
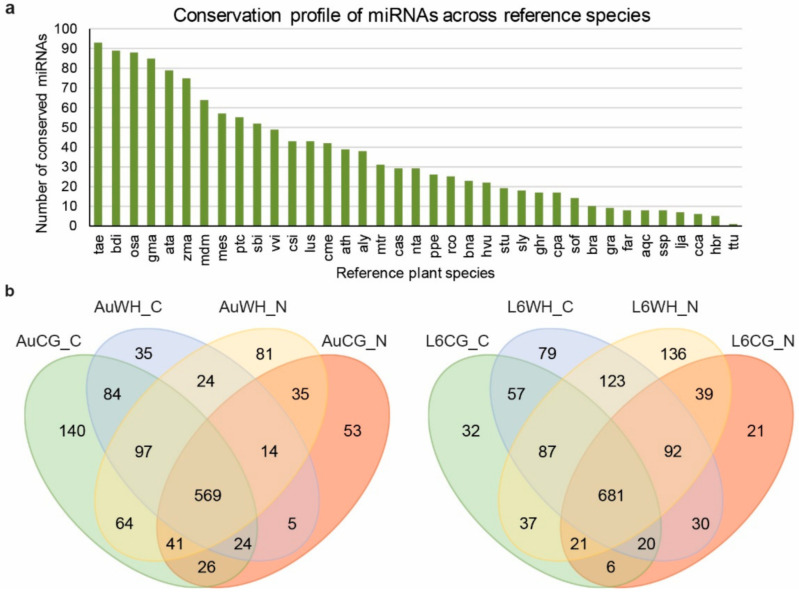
The conservation profile of durum wheat miRNAs across reference plant species (**a**) and the distribution of miRNAs across different treatment groups (**b**). The abbreviations of treatment groups are: AuCG_C (DBA Aurora control parents, progeny treated with the control), AuCG_N (DBA Aurora control parents, progeny treated with N starvation), AuWH_C (DBA Aurora water-deficit and heat stress parents, progeny treated with the control), AuWH_N (DBA Aurora water-deficit and heat stress parents, progeny treated with N starvation), L6CG_C (L6 control parents, progeny treated with the control), L6CG_N (L6 control parents, progeny treated with N starvation), L6WH_C (L6 water-deficit and heat stress parents, progeny treated with the control), and L6WH_N (L6 water-deficit and heat stress parents, progeny treated with N starvation). The abbreviations of reference species are: tae, *Triticum aestivum*. bdi, *Brachypodium distachyon*. osa, *Oryza sativa*. gma, *Glycine max*. ata, *Aegilops tauschii*. zma, *Zea mays*. mdm, *Malus domestica*. mes, *Manihot esculenta*. ptc, *Populus trichocarpa*. sbi, *Sorghum bicolor*. vvi, *Vitis vinifera*. csi, *Citrus sinensis*. lus, *Linum usitatissimum*. cme, *Cucumis melo*. ath, *Arabidopsis thaliana*. aly, *Arabidopsis lyrata*. mtr, *Medicago truncatula*. cas, *Camelina sativa*. nta, *Nicotiana tabacum*. ppe, *Prunus persica*. rco, *Ricinus communis*. bna, *Brassica napus*. hvu, *Hordeum vulgare*. stu, *Solanum tuberosum*. sly, *Solanum lycopersicum*. ghr, *Gossypium hirsutum*. cpa, *Carica papaya*. sof, *Saccharum officinarum*. bra, *Brassica rapa*. gra, *Gossypium raimondii*. far, *Festuca arundinacea*. aqc, *Aquilegia caerulea*. ssp., *Saccharum* ssp. lja, *Lotus japonicas*. cca, *Cynara cardunculus*. hbr, *Hevea brasiliensis*. ttu, *Triticum turgidum*.

**Figure 2 plants-10-00826-f002:**
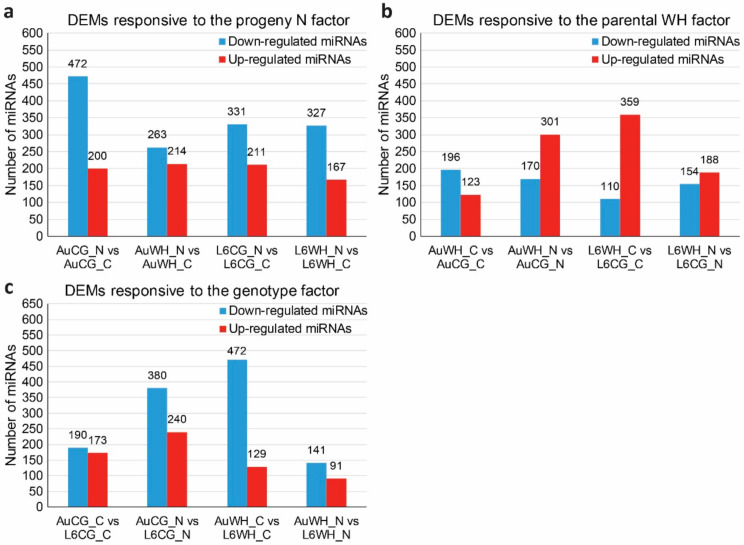
Number of differentially expressed miRNAs (DEMs) in response to difference factors. (**a**) DEMs responsive to the progeny N starvation factor. (**b**) DEMs responsive to the parental WH (water-deficit and heat stress) factor. (**c**) DEMs responsive to the genotype factor. The abbreviations of treatment groups are: AuCG_C (DBA Aurora control parents, progeny treated with the control), AuCG_N (DBA Aurora control parents, progeny treated with N starvation), AuWH_C (DBA Aurora water-deficit and heat stress parents, progeny treated with the control), AuWH_N (DBA Aurora water-deficit and heat stress parents, progeny treated with N starvation), L6CG_C (L6 control parents, progeny treated with the control), L6CG_N (L6 control parents, progeny treated with N starvation), L6WH_C (L6 water-deficit and heat stress parents, progeny treated with the control), and L6WH_N (L6 water-deficit and heat stress parents, progeny treated with N starvation).

**Figure 3 plants-10-00826-f003:**
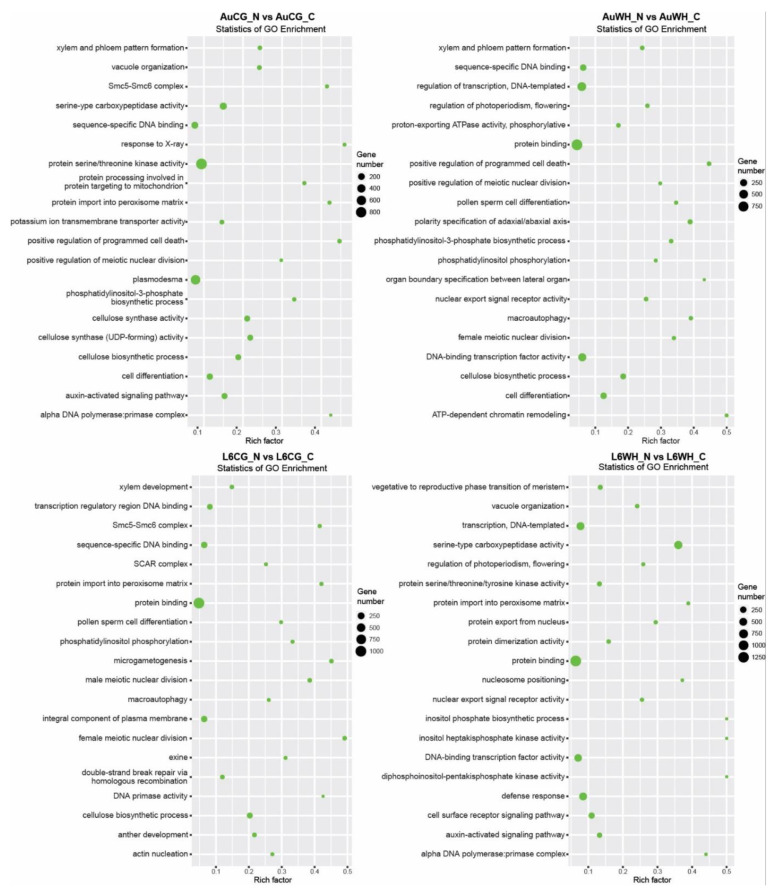
GO enrichment scatterplots of gene transcripts targeted by miRNAs responsive to the progeny N starvation factor. The treatment groups are: AuCG_C (DBA Aurora control parents, progeny treated with the control), AuCG_N (DBA Aurora control parents, progeny treated with N starvation), AuWH_C (DBA Aurora water-deficit and heat stress parents, progeny treated with the control), AuWH_N (DBA Aurora water-deficit and heat stress parents, progeny treated with N starvation), L6CG_C (L6 control parents, progeny treated with the control), L6CG_N (L6 control parents, progeny treated with N starvation), L6WH_C (L6 water-deficit and heat stress parents, progeny treated with the control), and L6WH_N (L6 water-deficit and heat stress parents, progeny treated with N starvation). Scatter dot size represents the number of targets annotated with a specific GO term. Rich factor represents the degree of enrichment (higher rich factor represents higher degree of gene enrichment).

**Figure 4 plants-10-00826-f004:**
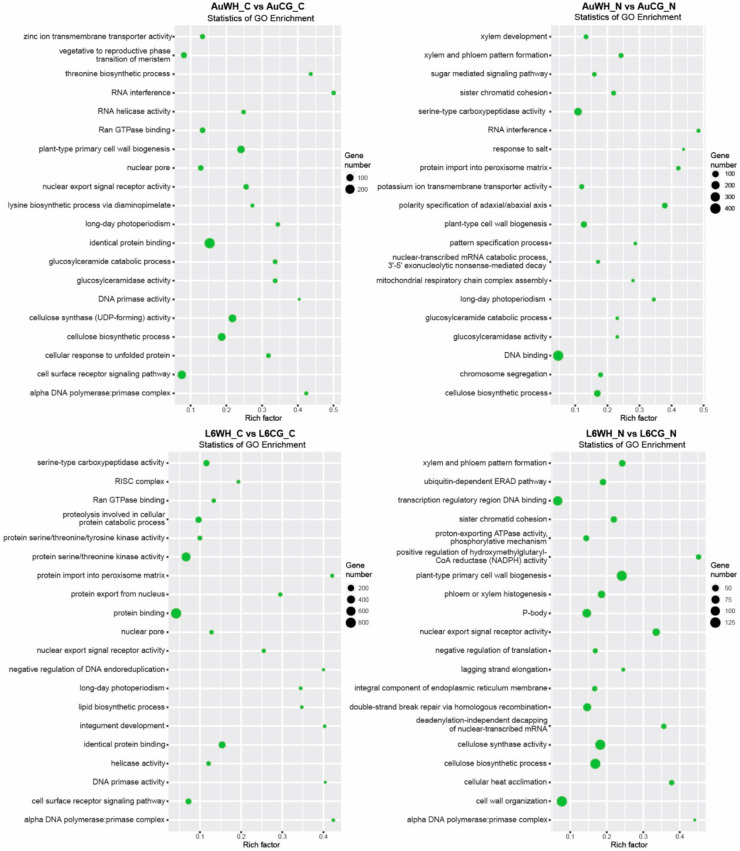
GO enrichment scatterplot of gene transcripts targeted by miRNAs responsive to the transgenerational effects of parental stress treatment. The treatment groups are: AuCG_C (DBA Aurora control parents, progeny treated with the control), AuCG_N (DBA Aurora control parents, progeny treated with N starvation), AuWH_C (DBA Aurora water-deficit and heat stress parents, progeny treated with the control), AuWH_N (DBA Aurora water-deficit and heat stress parents, progeny treated with N starvation), L6CG_C (L6 control parents, progeny treated with the control), L6CG_N (L6 control parents, progeny treated with N starvation), L6WH_C (L6 water-deficit and heat stress parents, progeny treated with the control), and L6WH_N (L6 water-deficit and heat stress parents, progeny treated with N starvation). Scatter dot size represents the number of targets annotated with a specific GO term. Rich factor represents the degree of enrichment (higher rich factor represents higher degree of gene enrichment).

**Figure 5 plants-10-00826-f005:**
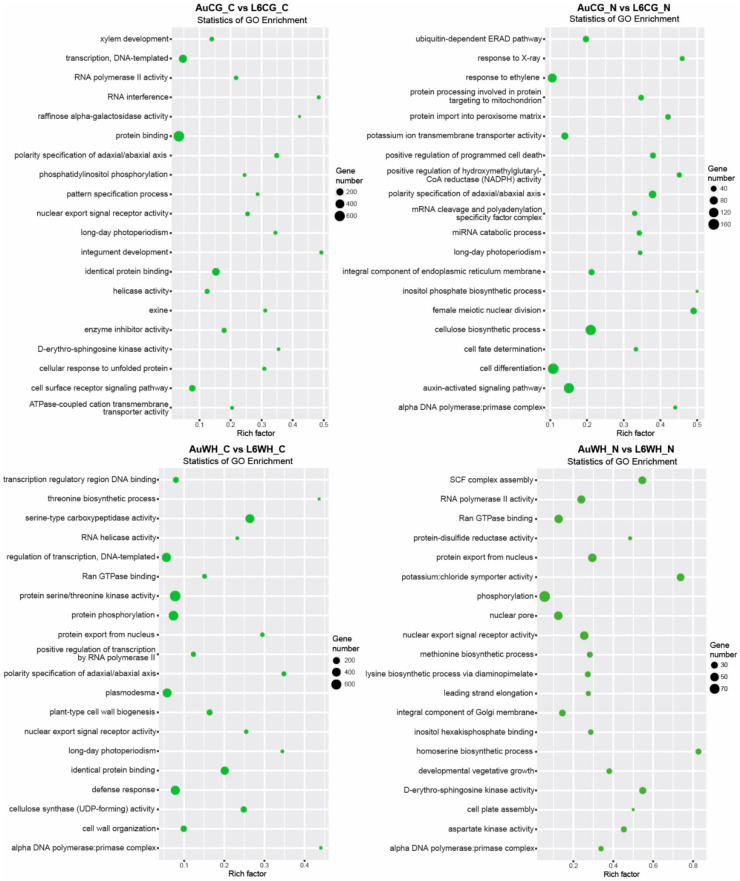
GO enrichment scatterplot of gene transcripts targeted by miRNAs with genotypic expression pattern. The treatment groups are: AuCG_C (DBA Aurora control parents, progeny treated with the control), AuCG_N (DBA Aurora control parents, progeny treated with N starvation), AuWH_C (DBA Aurora water-deficit and heat stress parents, progeny treated with the control), AuWH_N (DBA Aurora water-deficit and heat stress parents, progeny treated with N starvation), L6CG_C (L6 control parents, progeny treated with the control), L6CG_N (L6 control parents, progeny treated with N starvation), L6WH_C (L6 water-deficit and heat stress parents, progeny treated with the control), and L6WH_N (L6 water-deficit and heat stress parents, progeny treated with N starvation). Scatter dot size represents the number of targets annotated with a specific GO term. Rich factor represents the degree of enrichment (higher rich factor represents higher degree of gene enrichment).

**Figure 6 plants-10-00826-f006:**
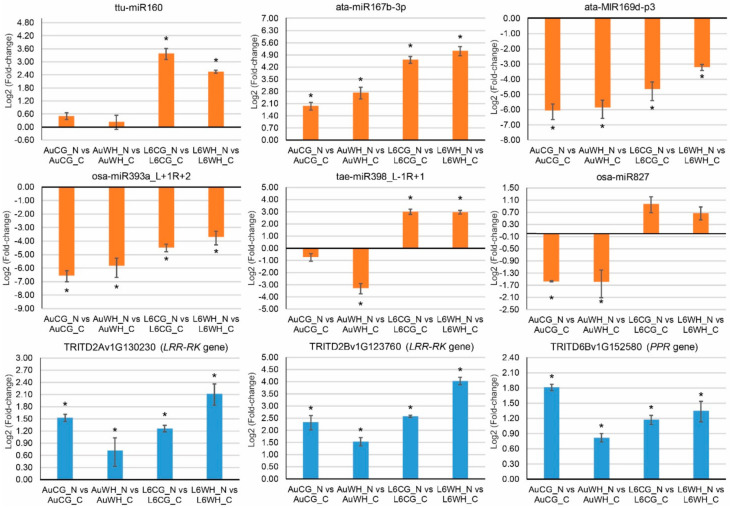
qPCR analysis of six N stress responsive miRNAs and three target genes. Two leucine-rich repeat receptor-like protein kinase family protein genes (*LRR-PK*) are targeted by ata-MIR169d-p3, and the pentatricopeptide repeat-containing protein gene (*PPR*) is targeted by osa-miR393a_L+1R+2. The treatment groups are: AuCG_C (DBA Aurora control parents, progeny treated with the control), AuCG_N (DBA Aurora control parents, progeny treated with N starvation), AuWH_C (DBA Aurora water-deficit and heat stress parents, progeny treated with the control), AuWH_N (DBA Aurora water-deficit and heat stress parents, progeny treated with N starvation), L6CG_C (L6 control parents, progeny treated with the control), L6CG_N (L6 control parents, progeny treated with N starvation), L6WH_C (L6 water-deficit and heat stress parents, progeny treated with the control), and L6WH_N (L6 water-deficit and heat stress parents, progeny treated with N starvation). Relative gene expression was calculated using GAPDH as the reference gene. Results are presented as log2 (fold-change) based on the relative gene expression of three biological replicates. Student’s t-test was used to detect statistical significance at *p* < 0.05 (indicated by *) for each pairwise comparison.

**Table 1 plants-10-00826-t001:** Seedling performance traits measured in DBA Aurora. The treatment groups are: AuCG_C (DBA Aurora control parents, progeny treated with the control), AuCG_N (DBA Aurora control parents, progeny treated with N starvation), AuWH_C (DBA Aurora water-deficit and heat stress parents, progeny treated with the control), AuWH_N (DBA Aurora water-deficit and heat stress parents, progeny treated with N starvation). Results shown as mean ± SE (*n* = 6).

Treatment Group	Seedling Height (cm)	Leaf Number	Shoot Fresh Weight (g)	Shoot Dry Weight (g)	Root Fresh Weight (g)	Root Dry Weight (g)	Primary Root Length (cm)	Chlorophyll Content (SPAD Units)
AuCG_C	36.92 ± 0.77	5.42 ± 0.15	1.658 ± 0.058	0.211 ± 0.009	1.327 ± 0.026	0.155 ± 0.004	27.62 ± 0.46	48.33 ± 0.70
AuCG_N	24.17 ± 0.42	3.17 ± 0.25	0.605 ± 0.022	0.087 ± 0.003	0.779 ± 0.018	0.094 ± 0.001	23.80 ± 0.34	38.38 ± 0.57
% Reduction	34.5%	41.5%	63.5%	60.7%	41.3%	39.3%	13.8%	20.6%
AuWH_C	35.93 ± 0.53	5.33 ± 0.17	1.630 ± 0.060	0.214 ± 0.006	1.328 ± 0.027	0.154 ± 0.003	28.77 ± 0.34	49.18 ± 0.55
AuWH_N	24.13 ± 0.50	3.25 ± 0.17	0.640 ± 0.019	0.089 ± 0.001	0.804 ± 0.023	0.097 ± 0.002	25.52 ± 0.41	40.28 ± 0.55
% Reduction	32.8%	39.1%	60.7%	58.4%	39.4%	37.1%	11.3%	18.1%
F pr. Parent treatment	0.381	1.000	0.940	0.667	0.584	0.876	0.001	0.032
F pr. Progeny treatment	<0.001	<0.001	<0.001	<0.001	<0.001	<0.001	<0.001	<0.001
F pr. Parent × Progeny treatment	0.413	0.663	0.481	0.412	0.613	0.447	0.476	0.388
l.s.d Parent treatment	n.a ^1^	n.a	n.a	n.a	n.a	n.a	0.813	1.242
l.s.d Progeny treatment	1.184	0.393	0.092	0.011	0.049	0.005	0.813	1.242
l.s.d Parent × Progeny treatment	n.a	n.a	n.a	n.a	n.a	n.a	n.a	n.a

^1^ n.a, not applicable.

**Table 2 plants-10-00826-t002:** Seedling performance traits measured in L6. The treatment groups are: L6CG_C (L6 control parents, progeny treated with the control), L6CG_N (L6 control parents, progeny treated with N starvation), L6WH_C (L6 water-deficit and heat stress parents, progeny treated with the control), L6WH_N (L6 water-deficit and heat stress parents, progeny treated with N starvation). Results shown as mean ± SE (*n* = 6).

Treatment Group	Seedling Height (cm)	Leaf Number	Shoot Fresh Weight (g)	Shoot Dry Weight (g)	Root Fresh Weight (g)	Root Dry Weight (g)	Primary Root Length (cm)	Chlorophyll Content (SPAD Units)
L6CG_C	35.47 ± 0.45	5.00 ± 0.22	1.556 ± 0.040	0.205 ± 0.005	1.313 ± 0.024	0.152 ± 0.002	27.13 ± 0.35	47.33 ± 0.73
L6CG_N	22.73 ± 0.57	2.83 ± 0.17	0.562 ± 0.010	0.076 ± 0.002	0.755 ± 0.019	0.090 ± 0.002	23.28 ± 0.37	35.97 ± 0.60
% Reduction	35.9%	43.3%	63.9%	63.1%	42.5%	40.9%	14.2%	24.0%
L6WH_C	34.42 ± 0.57	5.00 ± 0.18	1.502 ± 0.046	0.199 ± 0.006	1.303 ± 0.021	0.152 ± 0.003	27.83 ± 0.31	46.28 ± 0.56
L6WH_N	22.60 ± 0.63	2.75 ± 0.11	0.517 ± 0.011	0.072 ± 0.001	0.735 ± 0.022	0.088 ± 0.002	24.00 ± 0.31	34.90 ± 0.62
% Reduction	34.3%	45.0%	65.6%	63.9%	43.6%	41.7%	13.8%	24.6%
F pr. Parent treatment	0.301	0.815	0.133	0.211	0.507	0.704	0.047	0.108
F pr. Progeny treatment	<0.001	<0.001	<0.001	<0.001	<0.001	<0.001	<0.001	<0.001
F pr. Parent × Progeny treatment	0.421	0.815	0.893	0.759	0.826	0.799	0.980	0.990
F pr. Parent treatment	n.a ^1^	n.a	n.a	n.a	n.a	n.a	0.697	n.a
F pr. Progeny treatment	1.164	0.367	0.066	0.009	0.045	0.005	0.697	1.313
F pr. Parent × Progeny treatment	n.a	n.a	n.a	n.a	n.a	n.a	n.a	n.a

^1^ n.a, not applicable.

**Table 3 plants-10-00826-t003:** KEGG pathway enrichment analysis of the targets of N starvation-responsive DEMs. S gene number: the number of significant DEM target genes that match to the specified KEGG pathway; TS gene number: the number of significant DEM target genes that have annotated KEGG pathways; B gene number: the number of DEM target genes that match to the specified KEGG pathway; TB gene number: the number of DEM target genes that have annotated KEGG pathways. The abbreviations of treatment groups are: AuCG_C (DBA Aurora control parents, progeny treated with the control), AuCG_N (DBA Aurora control parents, progeny treated with N starvation), AuWH_C (DBA Aurora water-deficit and heat stress parents, progeny treated with the control), AuWH_N (DBA Aurora water-deficit and heat stress parents, progeny treated with N starvation), L6CG_C (L6 control parents, progeny treated with the control), L6CG_N (L6 control parents, progeny treated with N starvation), L6WH_C (L6 water-deficit and heat stress parents, progeny treated with the control), and L6WH_N (L6 water-deficit and heat stress parents, progeny treated with N starvation).

Pathway ID	KEGG Pathway Description	S Gene Number	TS Gene Number	B Gene Number	TB Gene Number
**AuCG_N vs. AuCG_C**					
ko04010	MAPK signaling pathway	286	1783	2171	50,492
ko04620	Toll-like receptor signaling pathway	167	1783	1298	50,492
ko04624	Toll and Imd signaling pathway	126	1783	633	50,492
ko05145	Toxoplasmosis	118	1783	1046	50,492
ko04621	NOD-like receptor signaling pathway	104	1783	1111	50,492
**AuWH_N vs. AuWH_C**					
ko04010	MAPK signaling pathway	101	1556	2171	50,492
ko04070	Phosphatidylinositol signaling system	91	1556	1224	50,492
ko04141	Protein processing in endoplasmic reticulum	89	1556	2406	50,492
ko00562	Inositol phosphate metabolism	84	1556	989	50,492
ko00230	Purine metabolism	84	1556	1483	50,492
**L6CG_N vs. L6CG_C**					
ko04141	Protein processing in endoplasmic reticulum	170	1451	2406	50,492
ko04070	Phosphatidylinositol signaling system	116	1451	1224	50,492
ko00562	Inositol phosphate metabolism	85	1451	989	50,492
ko00250	Alanine, aspartate and glutamate metabolism	73	1451	542	50,492
ko04144	Endocytosis	72	1451	1720	50,492
**L6WH_N vs. L6WH_C**					
ko04010	MAPK signaling pathway	326	1905	2171	50,492
ko04620	Toll-like receptor signaling pathway	186	1905	1298	50,492
ko05145	Toxoplasmosis	126	1905	1046	50,492
ko04141	Protein processing in endoplasmic reticulum	125	1905	2406	50,492
ko04624	Toll and Imd signaling pathway	122	1905	633	50,492

**Table 4 plants-10-00826-t004:** KEGG pathway enrichment analysis of the targets of DEMs responsive to the transgenerational effects of parental treatment. S gene number: the number of significant DEM target genes that match to the specified KEGG pathway; TS gene number: the number of significant DEM target genes that have annotated KEGG pathways; B gene number: the number of DEM target genes that match to the specified KEGG pathway; TB gene number: the number of DEM target genes that have annotated KEGG pathways. The abbreviations of treatment groups are: AuCG_C (DBA Aurora control parents, progeny treated with the control), AuCG_N (DBA Aurora control parents, progeny treated with N starvation), AuWH_C (DBA Aurora water-deficit and heat stress parents, progeny treated with the control), AuWH_N (DBA Aurora water-deficit and heat stress parents, progeny treated with N starvation), L6CG_C (L6 control parents, progeny treated with the control), L6CG_N (L6 control parents, progeny treated with N starvation), L6WH_C (L6 water-deficit and heat stress parents, progeny treated with the control), and L6WH_N (L6 water-deficit and heat stress parents, progeny treated with N starvation).

Pathway ID	KEGG Pathway Description	S Gene Number	TS Gene Number	B Gene Number	TB Gene Number
**AuWH_C vs. AuCG_C**					
ko04141	Protein processing in endoplasmic reticulum	87	789	2406	50,492
ko04010	MAPK signaling pathway	77	789	2171	50,492
ko03015	mRNA surveillance pathway	70	789	1365	50,492
ko03040	Spliceosome	68	789	1911	50,492
ko04144	Endocytosis	60	789	1720	50,492
**AuWH_N vs. AuCG_N**					
ko04010	MAPK signaling pathway	80	1031	2171	50,492
ko04070	Phosphatidylinositol signaling system	64	1031	1224	50,492
ko00562	Inositol phosphate metabolism	56	1031	989	50,492
ko04146	Peroxisome	53	1031	1272	50,492
ko03013	RNA transport	53	1031	1670	50,492
**L6WH_C vs. L6CG_C**					
ko04010	MAPK signaling pathway	258	1429	2171	50,492
ko04620	Toll-like receptor signaling pathway	141	1429	1298	50,492
ko03040	Spliceosome	109	1429	1911	50,492
ko04624	Toll and Imd signaling pathway	101	1429	633	50,492
ko04621	NOD-like receptor signaling pathway	100	1429	1111	50,492
**L6WH_N vs. L6CG_N**					
ko03018	RNA degradation	85	1187	1274	50,492
ko04070	Phosphatidylinositol signaling system	82	1187	1224	50,492
ko04141	Protein processing in endoplasmic reticulum	73	1187	2406	50,492
ko03040	Spliceosome	65	1187	1911	50,492
ko03008	Ribosome biogenesis in eukaryotes	52	1187	871	50,492

**Table 5 plants-10-00826-t005:** KEGG pathway enrichment analysis of the targets of DEMs with genotypic expression pattern. S gene number: the number of significant DEM target genes that match to the specified KEGG pathway; TS gene number: the number of significant DEM target genes that have annotated KEGG pathways; B gene number: the number of DEM target genes that match to the specified KEGG pathway; TB gene number: the number of DEM target genes that have annotated KEGG pathways. The abbreviations of treatment groups are: AuCG_C (DBA Aurora control parents, progeny treated with the control), AuCG_N (DBA Aurora control parents, progeny treated with N starvation), AuWH_C (DBA Aurora water-deficit and heat stress parents, progeny treated with the control), AuWH_N (DBA Aurora water-deficit and heat stress parents, progeny treated with N starvation), L6CG_C (L6 control parents, progeny treated with the control), L6CG_N (L6 control parents, progeny treated with N starvation), L6WH_C (L6 water-deficit and heat stress parents, progeny treated with the control), and L6WH_N (L6 water-deficit and heat stress parents, progeny treated with N starvation).

Pathway ID	KEGG Pathway Description	S Gene Number	TS Gene Number	B Gene Number	TB Gene Number
**AuCG_C vs. L6CG_C**					
ko04010	MAPK signaling pathway	91	922	2171	50,492
ko00230	Purine metabolism	78	922	1483	50,492
ko04141	Protein processing in endoplasmic reticulum	76	922	2406	50,492
ko03015	mRNA surveillance pathway	75	922	1365	50,492
ko00240	Pyrimidine metabolism	68	922	1168	50,492
**AuCG_N vs. L6CG_N**					
ko04141	Protein processing in endoplasmic reticulum	196	1969	2406	50,492
ko04010	MAPK signaling pathway	123	1969	2171	50,492
ko03018	RNA degradation	94	1969	1274	50,492
ko00230	Purine metabolism	84	1969	1483	50,492
ko03040	Spliceosome	84	1969	1911	50,492
**AuWH_C vs. L6WH_C**					
ko04010	MAPK signaling pathway	284	1457	2171	50,492
ko04620	Toll-like receptor signaling pathway	145	1457	1298	50,492
ko04624	Toll and Imd signaling pathway	108	1457	633	50,492
ko04621	NOD-like receptor signaling pathway	108	1457	1111	50,492
ko03040	Spliceosome	101	1457	1911	50,492
**AuWH_N vs. L6WH_N**					
ko00240	Pyrimidine metabolism	66	693	1168	50,492
ko00230	Purine metabolism	66	693	1483	50,492
ko03020	RNA polymerase	46	693	469	50,492
ko00270	Cysteine and methionine metabolism	46	693	1355	50,492
ko02010	ABC transporters	42	693	1530	50,492

## Data Availability

The small RNA sequencing datasets generated in the current study have been deposited in the NCBI GEO database under the accession number GSE168094.

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
