# Peer review of "Nitrogen Starvation-Responsive MicroRNAs Are Affected by Transgenerational Stress in Durum Wheat Seedlings"

_plants, 2021, doi:10.3390/plants10050826_

Round 1

Reviewer 1 Report

In this manuscript, the author studied nitrogen starvation-responsive microRNAs that are affected by transgenerational stress in durum wheat seedlings. In this study, the response of durum wheat seedlings to N starvation under the transgenerational effects of WH stress was investigated in two genotypes. Both genotypes showed a significant reduction in seedling height, leaf number, shoot and root weight (fresh and dry), primary root length, and chlorophyll content under N starvation stress. However, in the WH stress-tolerant genotype, the % reduction of most traits was lower in progeny from the stressed parents than progeny from the control parents. Small RNA sequencing identified 1,534 microRNAs in different treatment groups. Differentially expressed microRNAs (DEMs) were characterized as subject to N starvation, parental stress, and genotype factors, with their target genes identified in silico. GO and KEGG enrichment analyses revealed the biological functions, associated with DEM-target modules in stress adaptation processes, that could contribute to the phenotypic differences observed between the two genotypes. The manuscript is very well written. For the betterment of the manuscript, I have few comments to make.

It is important to validate the expression of DEM. Please choose some candidates and validate the expression level.

L268 and L355 correct the preposition at the end of a sentence.

Reviewer 2 Report

Dear Authors,

regarding my opinion your study “Nitrogen starvation-responsive microRNAs are affected by transgenerational stress in durum wheat seedlings” is comprehensive, excellently planned and conducted research, bringing at the same time a detailed and extensive source of information and novelty. It was a great pleasure to read your manuscript. 

I have just one minor comment:

  1. If it is possible Figures 2., 3.,4. and 5. should be presented in higher resolution. 

Thank you for your excellent, interesting and extensive work!

Best regards.

Reviewer 3 Report

The study reported in this manuscript, focuses on the morphological, physiological and molecular changes of two Australian durum wheat seedlings genotypes, water and heat (WH) stress-tolerant and WH stress-sensitive, under the combined effect of parental water-deficit and heat stress and progeny N starvation stress. Authors found that in the WH stress tolerant genotype, the % of reduction of the most of the studied traits was lower in the progeny of stressed parents than from control parents. Furthermore, the authors studied how miRNA can be involved in N starvation response under the effect of transgenerational WH stress, performing a systematic analysis of the miRNA expression profile. They identified differentially expressed miRNA subject to different factors from pairwise comparisons between (i) control and N starvation treatment groups with the same parental origin, (ii) groups with the same treatment factor but originated from different parental sources and (iii) groups with the same progeny treatment and parental treatment of two different genotypes. Protein-coding genes targeted by the differential expressed miRNA detected in the two durum wheat genotypes, were also identified and subjected to Gene Ontology and KEGG analyses.

In the opinion of this reviewer, this work focuses on an interesting topic that deserves to be investigated in depth: transgenerational inheritance of response and tolerance to stress in plants. Moreover, it is also novel and hence, relevant, because the effects of cross-stress (water and heat stress in the parental plants and N starvation in the progeny) are studied for the first time in durum wheat. The paper is very well written and clear. Experiments are designed and conducted well. Data set are sufficient to support conclusions and the discussion is exhaustive. Nevertheless, I have some comments and concerns listed below, that should be addressed before acceptance of the manuscript for publication in Plants.

- Table 1 and 2. Units of chlorophyll content are missing.

- Figure 2 should be bigger to see better the information of the graphs.

- I wonder if in L6 genotype there are specifically enriched GO terms for transcripts targeted by miRNA response to the progeny N starvation factor.

- Line 342. AuCG_C should be L6WH_C.

- Indicate if there are (or not) GO terms specifically enriched for AuWH_N vs L6WH_N and AuCG_C vs L6CG_C.

- KEGG pathway enrichment analysis of the targets of DEMs with genotypic expression pattern. In addition to the ABC transporters pathway only enriched for AuWH_N vs L6WH_N (lines 358-359), other pathways specifically enriched in each pairwise comparisons should also be indicated. For instance, ko03020 RNA polymerase and ko00270 Cysteine and methionine metabolism was also only enriched for AuWH_N vs L6WH_N or ko03018 RNA degradation was only enriched for AuCG_N vs L6CG_N. I suggest to comment those pathways specifically enriched for each pairwise comparison.

- I have a question regarding the use of the term “transgenerational”. At least in animals, is widely accepted that transgenerational inheritance is the concept that traits can be passed on from parent to great-grandchildren (in case of pregnant female directly exposed to the stressor) or grandchildren (in case of a male directly exposed to the stressor). However, in this work, the progeny exposed to N starvation showing a memory of parental exposure to WH stress was the first generation obtained from the parents. Would not be more appropriated to use the term “intergenerational” instead of “transgenerational” as in animals?

Major concerns

- Lines 498-500. It can be read “The up-regulation of leucine-rich repeat receptor genes via lowered miR169 abundance likely contributes to the adaptation of durum wheat to N stress via promoted N stress signalling”. Hence, it would be worth to investigate by qRT-PCR the expression of some of LRR-R genes targeted by miR169 in order to experimentally confirm their up-regulation in response to miR169 down-regulation due to N stress.

- In line with my previous comment, it would also be interesting to check by qRT-PCR the expression of the PPR gene that authors point in the discussion (line 566) is a target of the osa-miR393a_L+1R+2. This would allow experimentally confirming PPR up-regulation likely due to down-regulation of osa-miR393a_L+1R+2 in the progeny of AuWH_N vs AuCG_N.

Round 2

Reviewer 1 Report

I am happy with the reply of the authors. The manuscript looks better than before and can be accepted in its current format.